# A Health-Impact Assessment of an Ergonomic Measure to Reduce the Risk of Work-Related Lower Back Pain, Lumbosacral Radicular Syndrome and Knee Osteoarthritis among Floor Layers in The Netherlands

**DOI:** 10.3390/ijerph20054672

**Published:** 2023-03-06

**Authors:** P. Paul F. M. Kuijer, Henk F. van der Molen, Steven Visser

**Affiliations:** Netherlands Center for Occupational Diseases, Department of Public and Occupational Health, Amsterdam Public Health Research institute, Amsterdam Movement Sciences, Amsterdam UMC, University of Amsterdam, 1105 AZ Amsterdam, The Netherlands

**Keywords:** lower back pain, osteoarthritis, knee, prevalence, workplace, exposure, musculoskeletal diseases, risk factors, prevention, construction industry

## Abstract

Sand–cement-bound screed floor layers are at risk of work-related lower back pain, lumbosacral radicular syndrome and knee osteoarthritis, given their working technique of levelling screed with their trunk bent while mainly supported by their hands and knees. To reduce the physical demands of bending of the trunk and kneeling, a manually movable screed-levelling machine was developed for floor layers in the Netherlands. The aim of this paper is to estimate the potential health gains of working with a manually movable screed-levelling machine on the risk of lower back pain (LBP), lumbosacral radicular syndrome (LRS) and knee osteoarthritis (KOA) compared to traditional working techniques. This potential health gain was assessed using the epidemiological population estimates of the Population Attributable Fraction (PAF) and the Potential Impact Fraction (PIF), combined with work-related risk estimates for these three disorders from systematic reviews. The percentage of workers exceeding these risk estimates was based on worksite observations among 28 floor layers. For LBP, 16/18 workers were at risk when using traditional working techniques, with a PAF = 38%, and for those using a manually movable screed-levelling machine, this was 6/10 with a PIF = 13%. For LRS, these data were 16/18 with a PAF = 55% and 14/18 with a PIF = 18%, and for KOA, 8/10 with a PAF = 35% and 2/10 with a PIF = 26%. A manually movable screed-levelling machine might have a significant impact on the prevention of LBP, LRS and KOA among floor layers in the Netherlands, and health-impact assessments are a feasible approach for assessing health gains in an efficient way.

## 1. Introduction

Worldwide, the construction industry is characterized by a high prevalence of musculoskeletal complaints [1]. A review by Umer et al., showed that in the construction industry, the type of musculoskeletal complaint with the highest one-year prevalence concerns the lower back, accounting for 51%—followed by the knee, at 37%, and in third place the shoulder, at 32% [1]. The prevalence of other body regions is 30% for the wrist, 24% for the neck and ankle/foot, 20% for the elbow and upper back and 15% for the hip/thigh [1]. This prevalence of lower back and knee complaints is also high when looking at clinically assessed diagnoses of musculoskeletal diseases and disorders among construction workers. Dale et al., reported the annual prevalence of claims for acute musculoskeletal injuries (ICD10:S00-T14) and chronic musculoskeletal disorders (ICD10:M.x (x = any number)) over the period of January 2015 to June 2018 [2]. The percentage reported for the back/torso was 30% and the runners-up were both the lower and upper extremities, with 15%, respectively. Similar results have been reported by Van der Molen et al., in their study on the incidence rates of occupational diseases in the Dutch construction sector for 2010–2014 [3]. These incidence rates were based on a dynamic prospective cohort of occupational physicians reporting to the Netherlands Center for Occupational Diseases. An occupational disease is defined as a clinically assessed diagnosis that is predominantly caused by work-related factors, according to the reporting occupational physician [3]; the annual incidence of lower back pain (ICD-10 code M545) was the highest, with 750 cases per 100,000 construction workers. For osteoarthritis including the knee (ICD-10 codes M159, M169, M179, M189, M199) and excluding the spine, this was 688 cases per 100,000 construction workers. Not only self-reported complaints of the lower back and knee and physician-diagnosed (occupational) diseases or disorders of the lower back and knee, but also surgically treated musculoskeletal diseases and disorders regarding the lower back and knee appear to be high among construction workers—such as lumbar disc herniation [4,5] and hip and knee osteoarthritis [6]. Construction workers with these musculoskeletal diseases or disorders are at increased risk of sick leave [7] and paid labor-force exit due to work disability [8]. An occupation within the construction industry where workers run an increased risk of lower back pain, lumbosacral radicular syndrome and knee osteoarthritis is the that of sand–cement-bound screed floor layers [9,10,11].

To gain insight into the efficacy of preventive measures to reduce the number of floor layers with such diseases or disorders, insight into the proportional reduction of the number of these diseases or disorders is needed for floor layers that are not or are less exposed to the physical demands of this type of work [12,13]. In recent years, several systematic reviews have assessed to what extent physical demands at work contribute to these multifactorial musculoskeletal diseases and disorders, such as lower back pain [14], lumbosacral radiculopathy syndrome [15] and knee osteoarthritis [16]. Insight into the attributable fraction not only provides insight into the number of work-related diseases or disorders that might potentially be prevented, but can also be used to estimate the potential health benefits of a specific preventive measure. In particular, lumbosacral radicular syndrome and knee osteoarthritis have a long latency period before symptom onset. Therefore, a controlled prevention study to assess the incidence of these musculoskeletal diseases is not only time consuming, but probably requires a large number of participants to secure enough new cases and statistical power.

An alternative might be to perform a health-impact assessment. The World Health Organisation [17] defines a health-impact assessment as ”a practical approach used to judge the potential health effects of a policy, program or project on a population, particularly on vulnerable or disadvantaged groups. Recommendations are produced for decision-makers and stakeholders, with the aim of maximizing the proposal’s positive health effects and minimizing its negative health effects”.

In the Netherlands, the Dutch Labor Inspectorate wanted to reduce exposure to bending of the trunk and kneeling among sand–cement-bound screed floor layers and thereby reduce the risk of lower back pain, lumbosacral radicular syndrome and knee osteoarthritis by stimulating the use of a manually movable screed-levelling machine (Figure 1). Compared to traditional working techniques (Figure 1a), this work can be performed in a more upright standing and walking position (Figure 1b). This recommendation of the Dutch Labor Inspectorate was based on two studies of Visser et al. [11,18]: The first study [11] assessed the physical work demands of the traditional working techniques of sand–cement-bound screed floor layers and of anhydrite-bound screed floor layers [11]. The second study [18] assessed the physical work demands of only sand–cement bound screed floor layers using two electrical screed-levelling machines—namely, a manually movable screed-levelling machine (Figure 1b) and a self-propelled machine. Based on these two studies, Visser et al. [11,18] concluded that a manually movable screed-levelling machine may help to reduce the high physical work demands on floor layers while working with traditional working techniques. However, the studies by Visser et al. [11,18] did not answer the question of how great the health benefits are for floor layers regarding the reduction of the risk of lower back pain, lumbosacral radicular syndrome and knee osteoarthritis. To overcome this research gap, this paper aims to assess what the potential health benefits are for lower back pain, lumbosacral radicular syndrome and knee osteoarthritis using a health-impact assessment. Given that the exposure to bending of the trunk and kneeling among sand–cement-bound screed floor layers using manually movable screed-levelling machines is lower than that of using traditional working techniques, we hypothesized that manually movable screed-levelling machines result in a reduction of the risk of lower back pain, lumbosacral radicular syndrome and knee osteoarthritis. However, the real-world potential effect size has yet to be established.

In summary, therefore, the research question is: how much health gain can be expected from working with a manually movable screed-levelling machine compared to traditional working techniques in order to prevent lower back pain, lumbosacral radicular syndrome and knee osteoarthritis among sand–cement-bound screed floor layers in The Netherlands?

## 2. Materials and Methods

### 2.1. Design and Population

To answer the research question, we calculated the Population Attributable Fraction and the Potential Impact Fraction. To do so, we used the data from the studies by Visser et al. [11,12,13,14,15,16,17,18] that described the exposure to the physical work demands of bending of the trunk and kneeling. These two papers described, in total, four working techniques: The first paper described workplace assessments among sand–cement-bound screed floor and among anhydrite-bound screed floor layers [11]. The sand–cement-bound screed floor layers used traditional working techniques (Figure 1a), and these data were used in the present study. The second paper described similar workplace assessments, but this time among sand–cement bound screed floor layers using two electrical screed-levelling machines—namely, a manually movable screed-levelling machine (Figure 1b) and a self-propelled machine. In this paper, we only used the data on manually movable screed-levelling machines. This manually movable screed weighs about 24 kg and is 2 m wide (Figure 1b). The manually movable screed-levelling machine can be pushed, pulled, lifted or carried in the desired direction during the levelling of the screed floor.

The exposure to the physical work demands of bending of the trunk and kneeling using traditional working techniques and manually movable screed-levelling machines was assessed by means of real-time observations of, in total, 28 male floor layers during regular working days—18 floor layers working with traditional working techniques and 10 floor layers working with a manually movable screed-levelling machine [11,18]. The mean and standard deviation of the age, body height, body weight and seniority of these 28 screed floor layers were 41 (11) years, 181 (8) cm, 86 (12) kg and 16 (12) years, respectively.

In addition, to assess the number of screed floor layers at risk of lower back pain, lumbosacral radicular syndrome and knee osteoarthritis, the exposure limits for bending of the trunk and kneeling—as reported in the systematic reviews with meta-analyses of Lötters et al., Kuijer et al., and Verbeek et al.,—were used [14,15,16]. These exposure limits are defined in the following Section 2.2.

### 2.2. Population Attributable Fraction

To answer the research question, first, the population attributable fractions (PAF) was calculated using Formula (1) [19,20]:PAF = *p* × (OR − 1)/[1 + *p* × (OR − 1)](1)

with *p* being the prevalence of workers at risk of lower back pain, lumbosacral radicular syndrome or knee osteoarthritis that are exposed to the work-related risk factor at stake. For lower back pain and lumbosacral radicular syndrome, the risk factor at stake is working for 30 min or more per workday with the trunk bent by more than 40° [14,15]. For osteoarthritis of the knee, the risk factor at stake is kneeling for 60 min or more per workday [16]. In this paper we used the odds ratio (OR) instead of the relative risk given that the prevalence of these diseases or disorders is relatively low [21].

As such, the PAF shows what percentage of lower back pain, lumbosacral radicular syndrome and knee osteoarthritis can be attributed to physical work-demands during the work of sand–cement-bound screed floor layers based on traditional working techniques.

To calculate the PAF for lower back pain and lumbosacral radicular syndrome, as has been said, the exposure limit was defined at working for 30 min per day with the trunk bent by more than 40°. The corresponding ORs were derived from the systematic reviews with meta-analyses of Lötters et al., and Kuijer et al. [14,15]. For lower back pain, the OR = 1.7 (95% Confidence Interval (95%CI) 1.4–2.0) [14,22] (Table 2 in [14]), and for lumbosacral radicular syndrome, the OR = 2.4 (95%CI 1.7–3.6) (Figure 2 in [15]). For knee osteoarthritis, as has been said, this was defined based on kneeling for 60 min per workday, with a corresponding OR = 1.7 (95%CI 1.4–2.1) (Figure 1 in [16]). These exposure limits were also based on the reporting guidelines of the Netherlands Center for Occupational Diseases [23].

The percentage of workers exceeding these exposure limits was based on worksite observations among 18 floor layers using traditional working techniques. The observations are described in detail in the papers of Visser et al. [11,18]. In short, the work demands—the duration of bending of the trunk by more than 40° and the time spent kneeling—were the real-time observations of, in total, three observers using Task Recording and Analysis on a computer system at their workplace [24]. Each floor layer was observed by one observer. The observer was trained in real-time observations with the help of video fragments of floor layers using traditional working techniques and working with manually movable screed-levelling machines. The intra-observer reliability for the main tasks and work demands was sufficient and the intra-class coefficient ranged from 0.7 to 1.0. This interclass coefficient was considered adequate for workplace observations [11,18].

### 2.3. Potential Impact Fraction

Based on the PAF, the potential impact fraction (PIF) is estimated as the proportional reduction in incidence due to a reduction in the exposure to physical work demands [25]. The PIF is calculated using Formula (2) [26]:PIF = (*p* − P′) × (IDR − 1)/(*p* × (IDR − 1) + 1)(2)

with *p* being the prevalence of workers at risk while working without an ergonomic intervention, P′ being the prevalence of workers at risk when working with an ergonomic intervention and IDR being the Incidence Density Ratio—which in the present study was replaced with the OR.

The percentage of workers exceeding these exposure limits while working with a manually movable screed-levelling machine was based on worksite observations among 10 floor layers working with manually movable screed-levelling machines. These observations are described in detail in the papers of Visser et al., and a summary is given above in Section 2.1 [11,18].

## 3. Results

### 3.1. Population Attributable Fraction

The mean time working with the trunk bent by more than 40° was 98 min per worker per working day using the traditional working techniques of sand–cement-bound screed floor layers (Table 3 in [11]), and 16/18 workers were at risk of both lower back pain and lumbosacral radicular syndrome. For knee osteoarthritis, these data were 97 min per working day and 14/18 workers (Table 3 in [11]). This means that the maximum preventable work-related fraction based on the PAF for lower back pain was 38%, 55% for lumbosacral radicular syndrome and 35% for knee osteoarthritis (Figure 2).

### 3.2. Potential Impact Fraction

The mean time working with the trunk bent by more than 40° was 37 min per worker per working day when using a manually movable screed-levelling machine [18], and 6/10 workers were at risk of both lower back pain and lumbosacral radicular syndrome. For knee osteoarthritis, these data were 2/10 workers and 37 min per worker per working day (Table 1 in [18]). This means that the maximum preventable work-related fraction based on the PIF for lower back pain was 13%, 18% for lumbosacral radicular syndrome and 26% for knee osteoarthritis (Figure 2). When comparing the percentages of the PIF with the PAF, this means that manually movable screed-levelling machines seem most effective for the prevention of work-related knee osteoarthritis (26/35, 74%), next most effective for prevention of lumbosacral radicular syndrome (18/35, 51%) and least effective for prevention of lower back pain (13/38, 34%).

## 4. Discussion

The main finding of this study is that working with a manually movable screed-levelling machine might result in a reduction in lower back pain, lumbosacral radicular syndrome and knee osteoarthritis among floor layers in the Netherlands compared to traditional working techniques. In addition, since nine out of ten floor layers found the manually movable screed-levelling machine to be usable in practice [18], the Dutch Labor Inspectorate advices the use of traditional working techniques only in areas smaller than 30 m^2^—given the size and weight of the manually movable screed-levelling machine, to avoid unnecessary lifting and carrying. Moreover, this study shows what the added value might be of a health-impact assessment as a practical and efficient approach for estimating the potential health benefits for three prevalent work-related musculoskeletal disorders based on the use of an ergonomic measure at a worksite, without having to perform a prospective intervention study with a large group of workers and a follow-up of several years.

### 4.1. Comparison with Other Studies and Prospects

Regarding the estimated efficacy of a manually movable screed-levelling machine, we have to consider the following caveat: For a manually movable screed-levelling machine, the estimated reduction per worker per working day is about 1 h for bending of the trunk and 1 h for the time kneeling, given that only one floor layer operates the machine [18]. However, floor layers often work in teams of a total of three workers: one works with the manually movable screed-levelling machine; another floor layer, as the hodman, distributes the sand–cement mixture on the floor; and a third-floor layer sets out the height of the floor by manually levelling the floor around the walls. In practice, workers might rotate jobs during or between days. Therefore, a manually movable screed-levelling machine might change the work demands of all three workers and might have a smaller effect than that estimated in the present study. However, especially given the large effect on the exposure reduction of kneeling, this reduction might be sufficient to reduce the risk of knee osteoarthritis for all three floor layers. This expectation is in line with the findings of the studies of Jensen and Friche on knee complaints [27,28,29]; working more often in an upright working posture already reduced the number of floor layers reporting knee pain after 3 months (28% vs. 6%) [27]. After two years, floor layers who used the new upright working technique less often had a doubled risk of reporting complaints daily or for more than 30 days during the previous 12 months (OR 2.46, 95% CI 1.03 to 5.83) or reporting locking of the knees (OR 2.89, 95% CI 1.11 to 7.5) [28,29]. Moreover, the reduction in moderate-to-severe knee pain was greatest if floor layers started to use the new working methods before they developed knee complaints (OR 2.7 95%CI 1.02–7.26) [28,29]. These studies and the present study are also good examples that changes in the so-called ‘individual working practice’ by using assistive devices also contribute to a reduction in work-related musculoskeletal knee disorders [30,31]. This is important for knee osteoarthritis, given the strong, worldwide increase in this disabling disease—especially among workers—and the relatively little attention that is given to the prevention of work-related risk factors [32,33,34]; this remains important for the highly prevalent work-related lower back pain [35].

We wanted to compare the efficacy of manually movable screed-levelling machines on the reduction of the risk of musculoskeletal disorders with other ergonomic interventions that have been implemented and assessed in the workplace; unfortunately, we were not able to find any other ergonomic studies on the prevention of musculoskeletal disorders that have used a health-impact assessment or PIF to assess efficacy. Regarding the use of health-impact assessments in ergonomic studies, we found no other studies in PubMed on 19 January 2023. We retrieved 926 results using “Health-impact assessment” as a Mesh term in the PubMed database. This search was combined with “Ergonomics” as a Mesh term, including 60,624 results. This combined search with AND only resulted in two papers: one paper described potential health effects based on telework in response to the spread of COVID-19, and the other study assessed activities of daily living in older and healthy adults [36,37]. As such, neither of these two studies reported on potential health benefits for musculoskeletal disorders due to ergonomic interventions. To be more certain, we also performed a search in PubMed using PIF. We retrieved 71 results using ‘Potential Impact Fraction’ on 27 February 2023. Again, none of the studies reported on potential health benefits for musculoskeletal disorders due to ergonomic interventions. Most studies addressed the impact of a risk factor or intervention on cancer [38,39]; examples are obesity, smoking, alcohol consumption, fruit and vegetable intake and physical activity. Other preventable diseases that were often studied were diabetes and cardiovascular diseases. This emphasizes the merits of the use of a health-impact assessment in the field of ergonomic intervention studies to prevent work-related musculoskeletal disorders, as was performed in the present study. Given the numerous studies performed on physical exposure assessments in ergonomics to prevent these work-related musculoskeletal disorders [30], we suggest that researchers and practitioners consider including a simple health-impact assessment to estimate the potential health benefits—in terms of a musculoskeletal disease or disorder—of preventive ergonomic measures more often.

### 4.2. Strengths and Limitations

A strength of the present paper is that a health-impact assessment, as performed in the present study, might be a relatively simple tool to bridge the gap between ergonomic prevention studies on exposure reduction and epidemiological studies on potential health benefits. By using a health-impact assessment, additional insight is given into the extent an actual reduction in exposure might mean in terms of a specific work-related or occupational disease or disorder. Additionally, for other musculoskeletal diseases or disorders than the three described in the present study, reviews are available to set clinically relevant exposure limits; examples are carpal tunnel syndrome [40], lateral epicondylitis [41], subacromial pain syndrome [42] and hip osteoarthritis [43]. Another strength is the actual measurement of exposure at the worksite to assess the proportion of workers exceeding these health-related exposure limits [11,18].

A limitation is that we performed no follow-up study to validate whether workers using manually movable screed-levelling machines more often are indeed less susceptible to lower back pain, lumbosacral radicular syndrome and knee osteoarthritis in the upcoming years—as, for instance, Jensen and Friche did with a two-year follow up [28,29]. Even given the latency period required for lumbosacral radicular syndrome and knee osteoarthritis to become symptomatic, this might be manageable using a worker-specific occupational health surveillance program in the Dutch construction industry [44]. An example of such a study being feasible is an evaluation that was performed of whether an informational campaign resulted in an increased use of ergonomic measures and, subsequently, resulted in fewer self-reported musculoskeletal complaints over a five-year time period [45]; in this evaluation, the questionnaire data on occupational health surveillance were retrieved twice from a large cohort of about 1000 Dutch carpenters and pavers—once in 2000 and once in 2005. Another limitation might be that the observation time for working with a manually movable screed-levelling machine in the study of Visser et al. [18] was extrapolated to an entire working day to compare the data with the results of Visser et al. [11]. Since the mean time spent working with bending of the trunk was 37 min and around the exposure limit of 30 min, the prevalence of workers at risk might be different if these observations were performed during a full working day—as in the study of Visser et al. [11]. Most likely, the time spent bending is mainly dependent on the type of floor; it can be expected that the prevalence of workers at risk is lower when working in larger, open spaces where a manually movable screed-levelling machine can be used, and will be higher when working in narrow corridors—given that the size of a manually movable screed-levelling machine is too big for these latter circumstances, and therefore workers will use traditional working techniques.

## 5. Conclusions

Based on a health-impact assessment and calculating the potential impact fraction using workplace observations regarding exposure to physical work demands, we showed that the use of a manually movable screed-levelling machine might have a significant impact on the prevention of lower back pain, lumbosacral radicular syndrome and knee osteoarthritis among floor layers in the Netherlands, compared to traditional working techniques. The estimated percentage reduction in the preventable work-related fraction varied between 74% for knee osteoarthritis and 34% for lower back pain. Moreover, this paper shows that a health-impact assessment is a relatively simple approach for estimating health benefits in ergonomic prevention studies on the prevention of musculoskeletal diseases and disorders.

## Figures and Tables

**Figure 1 ijerph-20-04672-f001:**
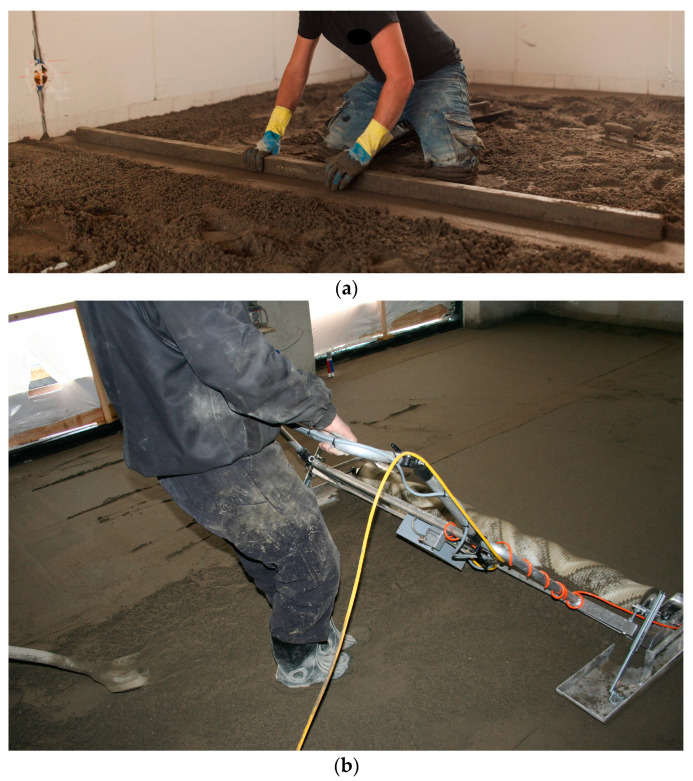
A sand–cement-bound screed floor layer working (**a**) using traditional working techniques and (**b**) using a manually movable screed-levelling machine.

**Figure 2 ijerph-20-04672-f002:**
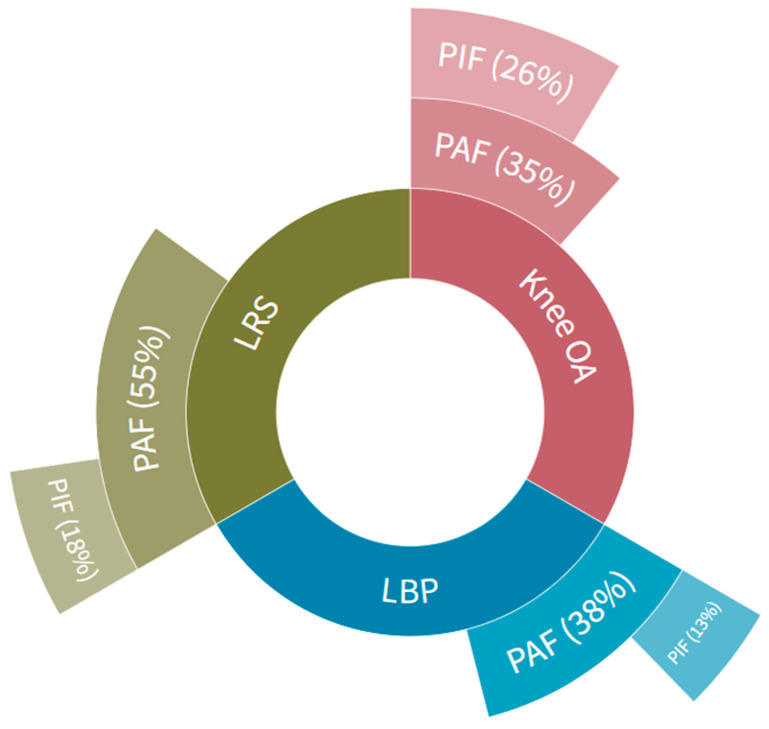
The maximum preventable work-related fraction (%) for lower back pain (LBP), for lumbosacral radicular syndrome (LRS) and for knee osteoarthritis (Knee OA) based on the Population Attributable Fraction (PAF) while working with traditional working techniques and the potential impact fraction (PIF, also in %) while working with manually movable screed-levelling machines.

## Data Availability

Not applicable.

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
