# Peer review of "A Health-Impact Assessment of an Ergonomic Measure to Reduce the Risk of Work-Related Lower Back Pain, Lumbosacral Radicular Syndrome and Knee Osteoarthritis among Floor Layers in The Netherlands"

_ijerph, 2023, doi:10.3390/ijerph20054672_

Round 1
Reviewer 1 Report
Dear authors, I present below my observations on the submitted manuscript:
Lines 107 to 109: I understand that there was an anticipation of the results found, when describing the characteristics of the evaluated subjects.
Equations could be numbered and presented more clearly.
Lines 147 and 148: Also understood as anticipating the result in the applied methods section.
Lines 232 to 242: I think there should be another objective in research and that is to survey the literature on similar methods and research. In the form presented in the discussion, the approach is dislocated and lost, without anchoring in a specific question.
Author Response
Dear authors, I present below my observations on the submitted manuscript:
- First of all, we would like to thank the reviewer for taking the time and effort to review our manuscript and for the helpful comments. Hopefully we have answered your comments in a satisfactory manner. The page lines refer to document with track changes.
Lines 107 to 109: I understand that there was an anticipation of the results found, when describing the characteristics of the evaluated subjects.
- We have added information on the workplace assessments and types of screed floor layers observed to clarify the use of both studies (lines 89-93, lines 125-135)
Equations could be numbered and presented more clearly.
- Thank you and we have followed up your suggestion and numbered both formulas for PAF and PIF, see lines 154 and 195.
Lines 147 and 148: Also understood as anticipating the result in the applied methods section.
- Thank you and we added more information about the two previously performed studies so it is now more clear that the observations were already performed and that we only use the outcomes in the present paper, see lines 125-144 and 158-161.
Lines 232 to 242: I think there should be another objective in research and that is to survey the literature on similar methods and research. In the form presented in the discussion, the approach is dislocated and lost, without anchoring in a specific question.
- Sorry for this misunderstanding. We thought it would be informative to show in the discussion in the paragraphs about ‘Comparison with other studies and prospects’ whether other ergonomic intervention studies have used a health impact assessment to evaluate the effects on musculoskeletal disorders. Then we could make a comparison with our findings regarding the efficacy. Unfortunately no other studies were found. To substantiate this finding we also provided the search strategy for the readers and updated the search. To be more certain, this time we also performed a search using Potential Impact Fraction. For transparency we also provided our search terms. To overcome the misunderstanding that we have an additional research question, we have reformulated the discussion paragraph, see lines 278-282 and 289-299. Hopefully this is now more clear.
Reviewer 2 Report
The article presents the results of an analysis of differences in construction workers' exposure to MSD's. The authors noted that reducing the necessary trunk bending and kneeling affects the decrease of disease symptoms. They presented the results based on numerous studies. However, the presentation of the results created several concerns, which I have presented in the comments below.
Page 2, lines: 87-88: The cited work presents tools different from those in Figure 1b. Please explain the differences in operation and describe the technical parameters.
Page 3, lines: 104-107: Was the study of 28 men published in article #11?
What parameters were observed? What observation tools were used - if TRAC, what variant?
I think it is necessary to characterize the research tools and methods in detail, since there is a wide variation in construction tools and techniques in the publications cited in Section 2.1. For example, working with cement screed is very different from pouring a self-leveling floor (as in publication 11).
Page 6, lines: 205-208: I think it is necessary to describe the performance differences using different tools.
Author Response
The article presents the results of an analysis of differences in construction workers' exposure to MSD's. The authors noted that reducing the necessary trunk bending and kneeling affects the decrease of disease symptoms. They presented the results based on numerous studies. However, the presentation of the results created several concerns, which I have presented in the comments below.
- First of all, we would like to thank the reviewer for taking the time and effort to review our manuscript and for the helpful comments. Hopefully we have answered your comments in a satisfactory manner. The page lines refer to document with track changes.
Page 2, lines: 87-88: The cited work presents tools different from those in Figure 1b. Please explain the differences in operation and describe the technical parameters.
- You are right. Sorry for the misunderstanding. The paper of Visser et al. [18] described two working techniques of sand–cement bound screed floor layers namely 1) using a manually moved screed levelling machine and 2) and a self-propelled machine. The manually moved screed levelling machine is depicted in figure 1b. The other paper of Visser et al. [11] described the traditional working technique of sand–cement-bound screed floor (and of anhydrite-bound screed floor layers). The traditional working technique of sand–cement-bound screed floor is depicted in figure 1a. We have amended the text in the introduction and in the methods section to clarify this, see lines 89-93 and lines 125-135.
Page 3, lines: 104-107: Was the study of 28 men published in article #11?
- These 28 men are based on the two studies of Visser et al. [11,18]: 18 floor layers while working with the traditional working technique [11] and 10 floor layers while working with the manually moved screed levelling machine [18]. We have amended the text accordingly, see lines 136-143.
What parameters were observed?
- Two observed parameters were used in the present study namely 1) the duration of bending of the trunk more than 40° and 2) the time kneeling, lines 136-138 and lines 158-161.
What observation tools were used - if TRAC, what variant?
- Real-time observations were used on the work site using TRAC (Task Recording and Analysis on Computer). There is only one TRAC variant and described in the following paper [24]:Frings-Dresen, M.H.W.; Kuijer, P.P.F.M. The TRAC-system: An observation method for analysing work demands at the workplace. Saf. Sci. 21(2), 163–165. https://doi.org/10.1016/0925-7535(95)00049-6
I think it is necessary to characterize the research tools and methods in detail, since there is a wide variation in construction tools and techniques in the publications cited in Section 2.1. For example, working with cement screed is very different from pouring a self-leveling floor (as in publication 11).
- Thank you again for pointing this out. We have tried to be more specific and therefore amended the text in the introduction, lines 89-93 and in the methods section, lines 125-135.. Please see also our reply to your second comment.
Page 6, lines: 205-208: I think it is necessary to describe the performance differences using different tools.
- We have added the following remark about the performance difference between the traditional working technique and the manually moved screed levelling machine: In addition, since nine out of ten floor layers found the manually moved screed levelling machine applicable in practice [18], the Dutch Labor Inspectorate advices to use the traditional working technique only in areas smaller than 30 m2 given the size and weight of the manually moved screed levelling machine to avoid unnecessary lifting and carrying, see lines 240-244.
Reviewer 3 Report
I have few concerns, which are mentioned below
1. In introduction section, author must include the research hypothesis and objectives which will further enhance the quality of manuscript. Furthermore, also include some research gap, and why the authors choose and focus on that study.
2. In materials and methods, why the authors choose the particular equation and what is the novelty of this methodology. Explain it thoroughly.
3. In discussion section, the significance of discussion is very low, the comparison with the other studies is not sufficient please also explain what is the contribution of the that study.
4. Conclusion is not well written, please rephrase it by including the brief description of your research.
5. The references are too small in quantity. Please include more latest references which will help the quality of manuscript.
Best regards
Author Response
Reviewer III
I have few concerns, which are mentioned below
- First of all, we would like to thank the reviewer for taking the time and effort to review our manuscript and for the helpful comments. Hopefully we have answered your comments in a satisfactory manner. The page lines refer to document with track changes.
- In introduction section, author must include the research hypothesis and objectives which will further enhance the quality of manuscript. Furthermore, also include some research gap, and why the authors choose and focus on that study.
- Thank you for this suggestion. We added the objective, the research gap and the hypothesis in the introduction, see lines 99-109.
- In materials and methods, why the authors choose the particular equation and what is the novelty of this methodology. Explain it thoroughly.
- Thank you for pointing this out. The methodology is not novel as can be seen from the references. The PAF reference [19] is from 2015 [and the PIF reference [26] from 1982. However, the novelty is that the present study is probably one of the first that uses a health impact assessment and calculates the PIF to estimate the efficacy of an ergonomic intervention based on the estimated reduction of three specific work-related musculoskeletal disorders. We felt that we could best address this topic in the discussion section, see lines . Hopefully this is acceptable for the reviewer.
- In discussion section, the significance of discussion is very low, the comparison with the other studies is not sufficient please also explain what is the contribution of the that study.
- We are in the opinion that the main contribution of this study is that we are one of the first to perform a health impact assessment and calculate the PIF to assess the efficacy of an ergonomic intervention on the risk reduction of musculoskeletal disorders. Until now this is not done in this field as far as we are aware of as is shown in the discussion section, see lines 278-282 and 289-301. This is in contrast with other fields, like for instance cancer and diabetes research.
- Conclusion is not well written, please rephrase it by including the brief description of your research.
- We have rephrased the conclusion by starting with a brief description of our research, see lines 342-344.
- The references are too small in quantity. Please include more latest references which will help the quality of manuscript.
- We have tried to include more recent references. Therefore we also performed an additional search on the use of the PIF in ergonomic studies on prevention of work-related musculoskeletal disorders, see lines 291-299. We included only two extra references from 2022 [44,45] given that we could not find any studies in the field of ergonomic intervention studies and the prevention of work-related musculoskeletal disorders.